# Mortality rate and predictors among stroke patients in the public hospitals in Harari region, Eastern Ethiopia

Alemayehu Tesfaye[1]*, Lemma Demissie Regassa[1], Birhanu Shegene[1], Nano Belema Areda[2], Assefa Tola[1]

1 School of Public Health, College of Health and Medical Sciences, Haramaya University, Harar, Ethiopia,
2 Department of Midwifery, College of Medicine and Health Science, Dire Dawa University, Dire Dawa, Ethiopia

* alextesfu3597@gmail.com

## Abstract

Stroke is a major cause of death and disability worldwide, yet there is limited information on the mortality rate and its predictors in Eastern Ethiopia. This lack of evidence is particularly significant, as hospitals in the Harari region provide the majority of healthcare services for stroke and chronic diseases. Therefore, our objective is to assess the mortality rate and predictors among stroke patients in the public hospitals in Harari. An institutional-based retrospective follow-up study was conducted among 452 randomly selected stroke patients at public hospitals in the Harari region from July 1, 2019, to June 30, 2024. The incidence of unfavorable treatment outcomes was calculated at 95% CI, and predictors of mortality were determined using Cox regression analyses. Of the 452 patients included, 292 (64.6%) improved, 21 (4.7%) were discharged with complications, 63 (13.9%) died, and 76 (16.8%) were discharged against medical advice. The 60-month follow-up revealed a mortality rate of 7.6 (95% CI: 5.9–9.7) per 1,000 person-months. The mortality risk was higher among stroke patients with hypertension (AHR: 2.0, 95% CI: 1.1-3.9), heart failure (AHR: 2.2, 95% CI: 1.1, 4.9), those with complications (AHR: 4.9, 95% CI: 1.5, 16.3), hospital-acquired infections (AHR: 3.1, 95% CI: 1.5-6.7), aspiration pneumonia (AHR: 1.9, 95% CI: 1.1-3.4), poor Glasgow Coma Scale (GCS) scores (AHR: 6.9, 95% CI: 2.4-19.9), and moderate impairment in GCS (AHR: 4.7, 95% CI: 1.6-13.3). Conversely, the use of antiplatelet drugs was associated with a reduced mortality risk in stroke patients (AHR: 0.5, 95% CI: 0.3-0.9). The mortality rate of stroke in this study was comparable to that of other studies in Ethiopia. Factors such as hypertension, heart failure, lower GCS, complications, aspiration pneumonia, and hospital-acquired infections increased mortality risks, while antiplatelet drugs reduced them. Therefore, strategies for early screening and follow-up of at-risk patients are essential.

**Data availability statement:** All data supporting the findings of this study are freely available as supplementary information.

**Funding:** This work was supported by Haramaya University (financial support to conduct the study to AT). The funders had no role in study design, data collection and analysis, decision to publish, or preparation of the manuscript.

**Competing interests:** The authors have declared that no competing interests exist.

## Introduction

Stroke is a serious public health concern that can occur suddenly and result in significant mortality and disability. It accounts for 11.6% of all mortality and 57% of Disability-Adjusted Life Years (DALYs), making it an important cause of mortality and disability globally. Between 1990 and 2019, there was a 43% rise in stroke-related mortality and a 32% increase in stroke-related DALYs [1]. Stroke is responsible for five to six million deaths globally each year; on average, one stroke-related mortality occurs every four minutes [2]. For patients with ischemic strokes, administering hypolipidemic and antiplatelet medications is associated with improved outcomes. Likewise, those experiencing hemorrhagic strokes tend to have better results when treated with hypolipidemic and antihypertensive medications [3]. Sixty percent of high-income countries (HICs) and 26% of nations with low and middle incomes offered acute stroke treatments [4]. Around 70% of mortality due to stroke and 87% of stroke-related disabilities happen in nations with low and middle incomes [5]. It's noteworthy that case fatalities appear to be rising in low and middle-income countries (LMICs) as opposed to HICs, underscoring the need for improved stroke care in LMICs [6]. Stroke inpatient mortality was high in sub-Saharan Africa. Stroke-related mortality was estimated to be 22% of the total population. In comparison to Eastern Africa (15%) and Southern Africa (18%), Western Africa had a higher stroke mortality rate (37%) [7].

Stroke fatalities among West Africans are linked to six patient characteristics: poor vegetable intake, elevated arterial pressure at presentation, more significant tumor volumes, elevated intracranial pressure (ICP), severe stroke, and aspiration pneumonia [8]. Population awareness, more intensive care unit (ICU) beds, telehealth, monitoring for late complications, and primary prevention, on the other hand, can lower in-hospital mortality rates [9]. Several conditions predicting mortality following an acute stroke were age, stroke type, stroke location, Glasgow Coma Scale degree of consciousness, NIHSS stroke severity, and comorbidities [9]. Stroke is associated with an increased risk of post-stroke infections [10], and they are a severe consequence for stroke patients in underdeveloped countries with insufficient rehabilitation services [11].

In Ethiopia, the severity of stroke and the need for better care are highlighted by the fact that almost one-fifth of stroke patients passed away while in the hospital [12]. The results of stroke treatment vary by region and over time. Stroke accounted for 7.5% to 19.3% of hospital admissions and 11% to 42.8% of fatalities from 2014 to 2019 [13]. Stroke-related complications were important indicators of death in Ethiopia [14].

Furthermore, stroke patients in Ethiopia typically have a dismal prognosis. Due to a shortage of computed tomography, excessively long prehospital delays, and a lack of essential drugs like r-tPA, most stroke patients experienced both neurologic and medical problems [15]. Data on stroke mortality are vital for tracking disease trends and organizing public health initiatives. Stroke mortality is a significant outcome metric in clinical trials and studies on stroke epidemiology. Effective management of

adult stroke patients requires the identification of mortality predictors. Yet, information regarding treatment outcomes and mortality predictors is insufficient in Harari Regional State. Therefore, assessing mortality rate and its predictions is the aim of the current study.

## Method and material

### Ethics statement

The study was carried out under consideration of the Helsinki Declaration of Medical Research Ethics [16]. This work has been approved by the Institutional Health Research Ethical Review Committee of the Haramaya University College of Health and Medical Sciences (Ref. No.. IHRERC/175/2024). Permission was obtained from the Haramaya University College of Health and Medical Science administration. The names of patients were not registered in the checklist, and their unique MRN numbers were locked for confidentiality. The need for written informed consent to participate was waived by the Institutional Health Research Ethical Review Committee of the Haramaya University College of Health and Medical Sciences due to the retrospective nature of the study. Data were accessed from July 15 to August 1, 2024.

### Study setting and design

An institutional-based retrospective follow-up study design was conducted among stroke patients at public hospitals in the Harari region, 526 km away from Ethiopia's capital city, Addis Ababa, from July 15 to August 1, 2024. There are currently two public, two private, one police, and one non-government hospital serving these individuals. In addition to the hospitals, the region's population is served by nine health facilities, twenty-nine private clinics, twenty-six health posts, and one regional laboratory. Jugal General Hospital (JGH) and Hiwot Fana Comprehensive Specialized University Hospital (HFC-SUH) are public hospitals in the Harari area that offer medical care to residents. Surgery, internal medicine, neurology services, mental health care, gynecology and obstetrics, pediatrics, maternal and child health (MCH), dental care, ophthalmology, TB and HIV (TB/HIV), intensive medical care, dermatology, and venereal disease services, pharmacy, oncologic services, and laboratory services are among health services they provide to the community [17].

### Populations and eligibility

All patients with stroke who were treated at public hospitals in Harari Regional State and whose age is greater than or equal to 15 years were the source population, whereas all stroke patients who were registered and admitted at the adult medical ward of public hospitals in the Harari region between July 1, 2019, and June 30, 2024, were our study population. All stroke patients greater than or equal to 15 years who were admitted to the medical ward in the study period were selected randomly and included in the study, whereas stroke patients with incomplete medical records about treatment outcome and an unknown date of diagnosis were excluded from the study.

### Sampling methods

The sample size for the first objective was calculated using the single population proportion formula by considering a similar study [18] with a 5% margin of error and a 95% confidence level. The sample size for the second objective (predictors of mortality of stroke patients) was calculated by considering the double population proportion formula and different factors that were significantly associated with the outcome variable, like types of strokes (hemorrhagic stroke) with assumed effect size AHR = 2.03 [19], Aspiration pneumonia AHR = 6.57 [20], Atrial fibrillation AHR = 1.104 [21], a two-sided confidence level of 95%, a margin of error of 5%, and a power of 80%, using EPI Info version 7 StatCalc software.

After comparing the results of the calculated sample size, a larger sample size of 434 was taken for this study. By adding a contingency of 10%, the final minimum sample size was 477 patients with stroke. A sampling frame was constructed by registering medical record numbers from the logbook of stroke patients. A total of 655 stroke patients (590 from

HFCSUH and 65 from Jugal General Hospital) were registered and treated between January 1, 2019, and June 30, 2024. Based on the number of stroke patients at each Hospital, a proportionate amount of a 477-person sample was assigned to each. Finally, the required number of participants was selected by simple random sampling using a computer-generated random sample from the sampling frame (430 patients were randomly selected from HFCSUH, and 47 patients were randomly selected from Jugal General Hospital).

## Data collection instrument and procedure

The data was collected using a data extraction format that was adapted from the WHO STEPSwise approach to stroke surveillance [22]. Data abstraction formats that contained relevant information about the patients, like demographic characteristics, outcomes of treatment, comorbidity, medication used for treatment, laboratory investigations, and clinical data, were used to abstract the data. Data were collected by four public health staff members of Haramaya University, and data collection was supervised by two trained supervisors. A patient's medical records (paper) were used to collect demographic characteristics, clinical data such as vital signs and types of strokes, comorbidity, medication for treatment, and outcomes of treatment.

## Variables

**Dependent variable.** In-hospital mortality.

**Independent variables.** Demographic factors (age, sex, residency)

Clinical-related factors (previous stroke, types of strokes, hypertension, DM, atrial fibrillation, aspiration pneumonia, raised intracranial pressure, Heart failure, kidney injury, dyslipidemia)

Treatment-related factors (lack of CT scan, prescribed medication, and the median time from onset of symptom to hospitalization).

## Operational definitions

**Treatment outcomes.** The results or effects of medical intervention on a patient's health (improved, complications, or death) [3].

**Improved.** Information about the improvement of patients was obtained from the discharge summary of medical records. Functional outcome was evaluated by the modified Rankin Scale (mRS). It was categorized into good (mRS < 3) and poor (mRS ≥ 3) functional recovery [23].

**Unfavorable treatment outcomes.** Negative results or effects that occur as a consequence of medical treatment [24]. These can include death, discharge with complications, and discharge against medical advice in this study.

**Good GCS.** Refers to a patient with a mild brain injury or who is alert (GCS 13–15); a moderate brain injury or who is drowsy (GCS 9–12); poor GCS is a patient with severe brain injury or who is unconscious (GCS 3–8) [25].

**Heart failure.** Diagnosed through a combination of patient history, physical examination, plus imaging-echocardiograms.

**Event.** A stroke patient who passed away from any cause while receiving treatment [13].

**Time to death.** Time from the date of stroke diagnosis to the date of death [13].

**Survival status.** The status of the patients' survival to the outcome (death) or censored [13].

## Data quality control

A pre-test was done in both hospitals for 5% of hospitalized stroke patients' medical records to ensure the reliability and variability of the data collection tools, and then all necessary adjustments were made to the data collection instruments. Data collectors were trained for a day on how to collect the data. Supervision was provided by the principal investigators during the data collection process, and any inconsistencies were amended on time.

## Methods of data analysis

The data collected by the Kobo tool was checked for consistency and completeness, sent to the server, imported, and analyzed by STATA software version 17.0. Descriptive statistics (mean, frequencies, tables, and graphs) were used to summarize and describe the data. A complete case analysis of a dataset with missing completely at random was used to deal with missing data. The cumulative incidence of mortality was calculated by taking the number of deaths as the numerator and the total initial population at risk on follow-up as the denominator. Patient-months at risk of mortality were calculated from the baseline diagnosis date to either the date of events or censoring. Accordingly, incidence density was computed as the number of deaths by patient-months at risk. The outcome variables were dichotomized into death (event) and censored. The Kaplan-Meier failure curve and a log-rank test were used to estimate the probability of mortality and to test the equality of failure functions among explanatory variables, respectively. Cox PH was fitted to identify the predictors of mortality. The Schoenfeld residuals test (both global and scaled) and graphical (log-log plot of survival) methods were used to check the proportional hazard (PH) assumption. The presence of multicollinearity was checked by using the variance inflation factor. Variables with a p-value of ≤0.2 were entered by stepwise regression into a multivariable Cox regression model to control for the possible effect of confounders. A P-value <0.05 was used to declare statistical significance in the multivariable model, and the hazard ratio (HR) with its 95% confidence interval will be computed to show the strength of the association.

## Results

### Socio-demographic and clinical characteristics of the Study Participants

A total of 477 records were screened, and 25(5.2%) patients with incomplete information were excluded (Fig 1). Of the total 452 study participants, the majority (90.0%) were from Hiwotfana Hospital; about two-thirds of them were males, 190 (42.0%) had ischemic stroke, and the mean age (±SD) was 55 years (±15.6). The most common clinical presentation was hemiparesis (76.8%), followed by loss of consciousness (32.3%), slurred speech (25.0%), and headache (24.3%). The median time of presentation from symptom onset was 36 (IQR: 54.0) hours, and about 53.5% of the time, symptom onset to admission was >24 hrs. The majority of them, 337 (74.6%), had comorbidity. Hypertension (55.3%) was the most common comorbid disease, followed by heart failure (11.7%), diabetes (11.1%), and kidney disease (9.3%). At hospitalization, the mean (SD) diastolic blood pressure and systolic blood pressure were 81.2±18.7 mmHg and 137.5±27.7 mmHg, respectively. About 8.2% of them had an elevated body temperature (>37.5°C). The mean total cholesterol was 145.4±48.9 mg/dl, and 13.6% of the patients had elevated total cholesterol levels; the median high-density lipoprotein (HDL) was 42±21.0. The median length of hospital stay was 5±6.0 days (Table 1).

### Treatment outcomes of stroke patients

From the total of 452 patients, 292 (64.6%) 95% CI (60.1%–68.9%) had improved outcomes, 21 (4.7%) 95% CI (3.0%–7.0%) were discharged with complications, about 63 (13.9%) 95% CI (11.0%–17.5%) died, and about 76 (16.8%) 95% CI (13.6%–20.6%) were discharged against medical advice on self and family request. About 160 (35.4%, 95% CI (31.1%–39.9%) had unfavorable outcomes. Of 63 patients who died during hospitalization, 31 (49.2%) had a hemorrhagic stroke. Aspiration pneumonia 37 (58.7%) and increased intracranial pressure 35 (55.6%) were the most frequently documented causes of death secondary to stroke. Of 160 unfavorable treatment outcomes that occurred during the follow-up period, more than half, 105 (65.6%), of them were males, and 83 (51.9%) were aged 45–65 years. Similarly, 111 (69.4%) had complications, and 94 (58.8%) were hypertensive at baseline. Among 452 patients, about 213 (47.1%) had complications. The most common complications during hospitalization were aspiration pneumonia (20.1%) and ICP/brain edema (20.4%). During hospitalization, the most commonly used antiplatelet and lipid-lowering drugs were Aspirin and Atorvastatin, administered to approximately 54.4% and 59.7% of the patients, respectively. Enalapril, which was administered

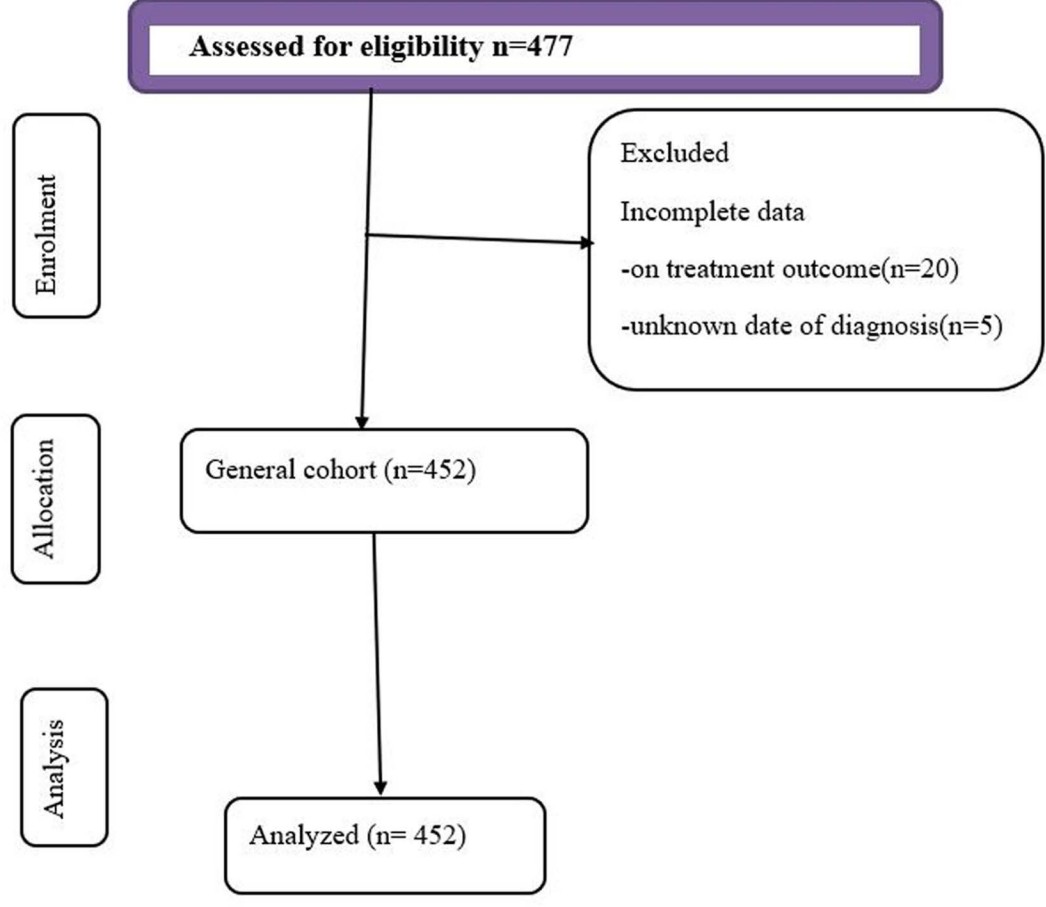

**Fig 1. Flow diagram of selection for study on mortality rate and its predictors among stroke patients at public hospitals in the Harari region, East Ethiopia, 2024.**

in around 29.4% of cases, was the antihypertensive drug that stroke patients used the most frequently. Ceftriaxone and metronidazole were administered to 28.5% of hospitalized stroke patients to treat aspiration pneumonia and sepsis, as well as to control the effects of stroke and comorbidities (Table 2).

## Overall survival rate of stroke patients

The participants were followed for a lowest of one day to a maximum of sixty months with no median survival time. Among the total stroke patients followed for approximately 60 months, 63 (13.9%, 95% CI (11.0%–17.5%) died, and 160 (35.4%, 95% CI (31.1%–39.9%) had unfavorable outcomes. The incidence of death was 7.6 cases per 1,000 person-months of observation with a 95% CI of (5.9–9.7). The cumulative probability of death among patients with stroke on the first day of admission was 2.5%; in the first month, it was 14.2%; in the third month, it was 14.7%; and from the fourth month to the end of the follow-up period, it was 15.4%. The incidence of unfavorable outcomes was 22.7 per 1,000 person-months of observation with a 95% CI of (19.4–26.5). The cumulative probability of unfavorable outcomes among patients with stroke

**Table 1. Socio-demographic and clinical characteristics of stroke patients at public hospitals in the Harari region, East Ethiopia, from 2019 to 2024. (N=452).**

| Characteristics | Category | Frequency (%) |
|---|---|---|
| Sex | Male | 300(66.4) |
| | Female | 152(33.6) |
| Age | <45 | 109 (24.1) |
| | 45-65 | 225 (49.8) |
| | >65 | 118 (26.1) |
| Residence | Rural | 299(66.2) |
| | Urban | 153(33.9) |
| Type of stroke | Ischemic | 190 (42.0) |
| | Hemorrhagic | 160 (35.4) |
| | Unclassified | 102 (22.6) |
| Clinical presentations | Hemiparesis | 347 (76.8) |
| | Loss of consciousness | 146 (32.3) |
| | Slurred speech | 113 (35.0) |
| | Headache | 110 (24.3) |
| | Aphasia | 93 (20.6) |
| | Vomiting | 67 (14.8) |
| | Facial palsy | 62 (13.7) |
| Previous history of stroke | No | 382 (84.5) |
| | Yes | 70 (15.5) |
| Time from onset to admission (in hrs.) | Median ±(IQR) =36±54.0 | |
| Comorbidity | Yes | 337 (74.6) |
| | No | 115 (25.4) |
| List of comorbidities | Hypertension | 250 (55.3) |
| | Heart failure | 53 (11.7) |
| | Diabetes | 50 (11.1) |
| | Kidney disease | 42 (9.3) |
| | Myocardial Infraction | 20 (4.4) |
| | Atrial fibrillation | 16 (3.5) |
| Systolic blood pressure (in mmHg) | Mean±(SD)= 137.46±27.7 | |
| Diastolic blood pressure (in mmHg) | Mean±(SD)=81.24±18.7 | |
| Respiratory rate (in breaths per minute) | 12-18 | 22 (4.9) |
| | >18 | 430 (95.1) |
| Pulse rate (in beats per minute) | <60 | 10 (2.2) |
| | 60-100 | 353 (78.1) |
| | >100 | 89 (19.7) |
| Body temperature(in°C) | <36.5 | 192 (42.5) |
| | 36.5-37.5 | 223 (49.3) |
| | >37.5 | 37 (8.2) |
| Random blood glucose (in mg/dl) | ≤200 | 425 (94.0) |
| | >200 | 27 (6.0) |
| Total cholesterol (mg/dl) | <200 | 336 (86.4) |
| | ≥200 | 53 (13.6) |
| Triglycerides(mg/dl) | <150 | 315 (81.4) |
| | ≥150 | 72 (18.6) |
| HDL (mg/dl) | Median ±(IQR) =42±21.0 | |

*(Continued)*

**Table 1.** (Continued)

| Characteristics | Category | Frequency (%) |
|---|---|---|
| LDL (mg/dl) | Median ±(IQR) =55.7±50.8 | |
| Serum creatinine (in mg/dl) | <0.5 | 35 (7.7) |
| | 0.5-1.2 | 354 (78.3) |
| | >1.2 | 63 (13.9) |
| Serum potassium level (in meq/l) | <3.5 | 84 (18.6) |
| | 3.5-5 | 342 (75.7) |
| | >5 | 26 (5.7) |
| Glasgow Coma Scale (GCS) | ·8 | 59 (13.1) |
| | 9-12 | 152 (33.6) |
| | 13-15 | 241 (53.3) |
| length of hospital stays(days) | Median ±(IQR) =5±6.0 | |

on the first day of admission was 5.1%; at 20.2 months, it was 33.1%; at 40.2 months, it was 40.3%; and at the end of the follow-up period, it was 45.4% (Fig 2).

Kaplan Meier failure curve and a log-rank test showed that the differences in cumulative probability of death of patients with hypertension and those without hypertension, with aspiration pneumonia and those without aspiration pneumonia were statistically significant (log-rank: $p=0.0017$, log-rank: $p=0.0001$, respectively) (Fig 3).

## Predictors of mortality

In the bi-variable Cox regression analysis, ten variables (age greater than 65 years, hypertension, heart failure, lower GCS, having complications, aspiration pneumonia, increased intracranial pressure, hospital-acquired infection, respiratory rate>18, and antiplatelet drug) were identified as the associated factors of mortality at p · 0.25. In the multivariable Cox regression analysis, seven variables were identified as predictors of mortality. These were hypertension, heart failure, lower GCS, having complications, aspiration pneumonia, hospital-acquired infection, and antiplatelet drugs.

Accordingly, hypertensive patients had a 2-fold higher probability of death because of stroke than non-hypertensive patients (AHR: 2.0, 95% CI: 1.1, 3.9). The risk of mortality of stroke was 2.2 times higher among patients with heart failure as compared to those without heart failure (AHR: 2.2, 95% CI: 1.1, 4.9). Having acute complications increased the hazard of death by 4.9 times compared to stroke patients who did not have complications (AHR: 4.9, 95% CI: 1.5, 16.3). The hazard of death from stroke was 3.1 times higher among patients with hospital-acquired infection as compared to those without hospital-acquired infection (AHR: 3.1, 95% CI: 1.5, 6.7). Stroke patients with aspiration pneumonia were 1.9 times more likely to die as compared to those without aspiration pneumonia (AHR: 1.9, 95% CI: 1.1, 3.4). The risk of mortality of stroke was 6.9 times higher among patients with GCS<8 as compared to those with GCS 13–15 (AHR: 6.9, 95% CI: 2.4, 19.9). The risk of experiencing mortality from stroke was 4.7 times higher among patients with GCS 9–12 as compared to those with GCS 13–15 (AHR: 4.7, 95% CI: 1.6, 13.3). Furthermore, the risk of mortality was 50.0% lower among stroke patients who took antiplatelet drugs as compared to those who didn't take antiplatelet drugs (AHR: 0.5, 95% CI: 0.3, 0.9) (Table 3).

## Discussion

This study included 452 patients with stroke (42.0%), 95% CI (37.6%–46.7%), who were diagnosed with ischemic stroke (35.4%), 95% CI (31.1%–39.9%), had hemorrhagic stroke, and (22.6%) 95% CI (19.0%–26.7%) patients with unclassified. Thirty-one patients (19.4%) with hemorrhagic stroke died, 18 patients with ischemic stroke (9.5%), and 14 patients

**Table 2. Treatment outcomes for stroke patients at public hospitals in the Harari region, East Ethiopia, from 2019 to 2024. (N = 452).**

| Characteristics | Category | | Frequency (%) |
|---|---|---|---|
| Treatment outcomes | Improved | | 292 (64.6) |
| | Unfavorable | Discharged with complications | 21 (4.7) |
| | | Died | 63 (13.9) |
| | | DAMA | 76 (16.8) |
| Complication | Yes | | 213 (47.1) |
| | No | | 239 (52.9) |
| Lists of complications | Brain edema/ICP | | 92 (20.4) |
| | Aspiration pneumonia | | 91 (20.1) |
| | UTI | | 36 (8.0) |
| | HAI | | 32 (7.1) |
| | Seizure | | 21 (4.7) |
| | Septic shock | | 19 (4.2) |
| Anti-platelet | Aspirin | | 246 (54.4) |
| | Clopidogrel | | 22 (4.9) |
| Anticoagulant | Heparin | | 188 (41.6) |
| | Warfarin | | 32 (7.1) |
| Statins | Atorvastatin | | 270 (59.7) |
| Anti-hypertensive | Enalapril | | 133 (29.4) |
| | Amlodipine | | 116 (25.7) |
| | Nifedipine | | 56 (12.4) |
| | Hydrochlorothiazide | | 46 (10.2) |
| | Captopril | | 16 (3.5) |
| Anti-ulcer drugs | Cimetidine | | 69 (15.3) |
| | Omeprazole | | 19(4.2) |
| Antibiotics | Ceftriaxone | | 118 (26.1) |
| | Metronidazole | | 11(2.4) |
| Anti-pyretic | Paracetamol | | 119 (26.3) |

with unclassified stroke (13.7%). The findings of this study showed that 63 (13.9%, 95% CI (11.0%–17.5%) of the stroke patients died, with an incidence density of 7.6 per 1,000 person-months (PM) of observation.

This study's cumulative incidence of mortality was in line with the study in Felege Hiwot Hospital, Ethiopia, 15.2% [13]; in North West Ethiopia, 14.5% [26]; in Ayder Comprehensive Specialized Hospital, Ethiopia, 14.9% [27]; in Debre Markos Comprehensive Specialized Hospital, Ethiopia, 12.8% [19]; in Gondar University Hospital, Ethiopia, 12.5% [28]. However, it was higher as compared to the study done by [29] at a comprehensive stroke care center in Kerala, India, 3.4%. On the other hand, the cumulative incidence in this study was lower than the study conducted in Sierra Leone (34.8%) [30], at University of Gondar Teaching Hospital, Tibebe Gion Comprehensive Specialized Hospital, and Felege Hiwot Referral Hospital, Ethiopia (27.1%) [31], at Mettu Karl Referral, Ethiopia (27.2%) [2], at Saint Paul's Hospital Millennium Medical College, Ethiopia (30.7%) [32], at Jimma University Medical Center, Ethiopia (51.6%) [18].

This could be due to differences in sample size, follow-up period, types of strokes, complications, and comorbidities. In Sierra Leone, 178 patients were followed for a maximum of 12 months, whereas this study followed 452 patients for a maximum of 60 months. In a study at the University of Gondar, Tibebe Gion Comprehensive Specialized Hospital, and Felege Hiwot Referral Hospital, Ethiopia, 57.9% of participants had ischemic stroke, compared to 42.0% in this study. At St. Paul's Hospital in Addis Abeba, 70.9% of 251 participants had hypertension, 43.8% had diabetes mellitus (DM), and

PLOS Global Public Health

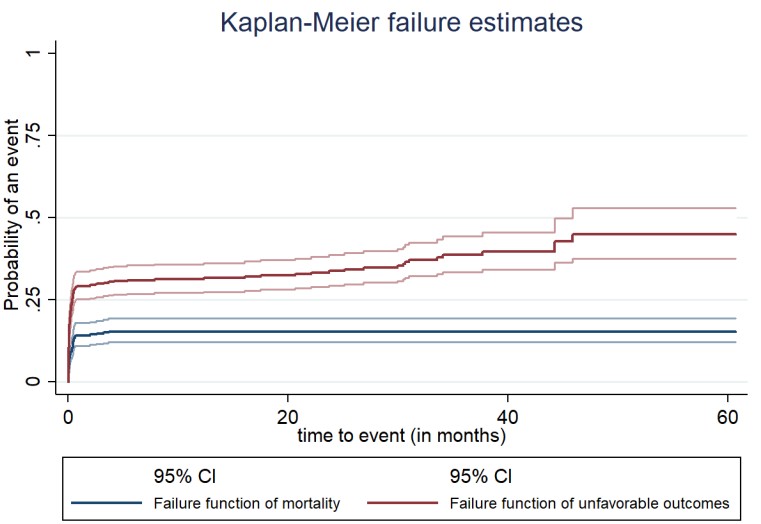

**Fig 2. Overall Kaplan-Meier curve for the cumulative probability of mortality and unfavorable outcomes of stroke patients at public hospitals in the Harari region, East Ethiopia, 2024.**

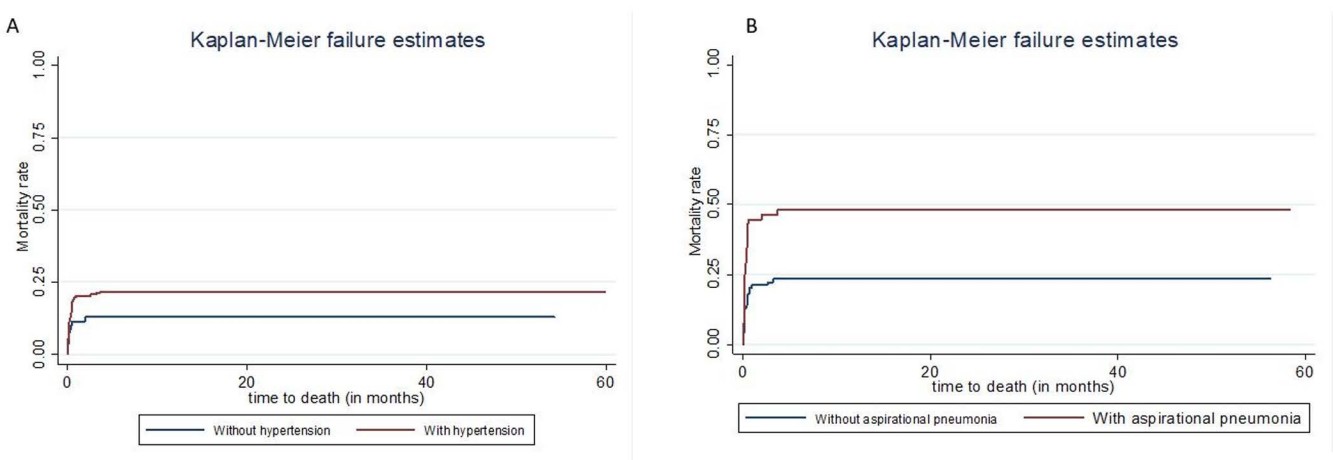

**Fig 3. Kaplan-Meier curve for mortality estimate among different groups of stroke patients by hypertension (A) and aspiration pneumonia (B) at public hospitals of Harari region, East Ethiopia, 2024.**

53.8% were hemorrhagic stroke patients. In contrast, 55.3% of 452 participants in this study had hypertension, 11.1% had DM, and 35.4% were hemorrhagic stroke patients. In Mettu, 56.9% of 202 participants had complications, while 52.9% of those in this study had no complications. In Jimma, 54% had ischemic stroke and 38% had DM, compared to 42.0% and 11.1%, respectively, in this study. The higher comorbidities and complications in other studies are associated with increased mortality. On the other hand, the Indian study used a prospective follow-up design.

Age older than 65 years was not statistically significant in the study. A related study conducted in North West Ethiopia found that stroke patients over the age of 65 had a 6.3 times higher risk of dying than those under the age of 45 [30]. This finding is congruent with several other studies, including those conducted in Beirut [33], Jimma University Medical

**Table 3. Bivariate and multivariate Cox regression analysis of predictors of mortality among stroke patients at public hospitals in the Harari region, East Ethiopia, 2024.**

| Variable | Category | Mortality | | CHR (95% CI) | AHR (95% CI) | P-value |
|---|---|---|---|---|---|---|
| | | Censored(n=389) | Event (n=63) | | | |
| Age | <45 | 99 | 10 | 1 | 1 | |
| | 45-65 | 202 | 23 | 0.9(0.5- 2.0) | 1.1(0.4-2.4) | 0.943 |
| | >65 | 88 | 30 | 3.0(1.4-6.1) | 1.6(0.7-3.6) | 0.232 |
| Residence | Rural | 254 | 45 | 1.2(0.7-2.1) | 1.1(0.6-1.9) | 0.948 |
| | Urban | 135 | 18 | 1 | 1 | |
| Hypertension | No | 188 | 14 | 1 | 1 | |
| | Yes | 201 | 49 | 2.5(1.4- 4.6) | 2.0(1.1- 3.9) | **0.033** |
| Heart Failure | No | 346 | 53 | 1 | 1 | |
| | Yes | 43 | 10 | 1.7(0.8-3.3) | 2.2(1.1-4.9) | **0.045** |
| GCS | ·8 | 24 | 35 | 28.7(11.1-73.8) | 6.9(2.4-19.9) | **0.001** |
| | 9-12 | 129 | 23 | 8.2(3.1-21.7) | 4.7(1.6-13.3) | **0.004** |
| | 13-15 | 236 | 5 | 1 | 1 | |
| Complication | No | 238 | 4 | 1 | 1 | |
| | Yes | 151 | 59 | 19.1(6.9-52.8) | 4.9(1.5-16.3) | **0.009** |
| Aspiration Pneumonia | No | 335 | 26 | 1 | 1 | |
| | Yes | 54 | 37 | 7.0(4.2-11.6) | 1.9(1.1- 3.4) | **0.038** |
| Increased ICP | No | 332 | 28 | 1 | 1 | |
| | Yes | 57 | 35 | 5.1(3.1-8.5) | 1.3(0.7-2.4) | 0.484 |
| HAI | No | 370 | 50 | 1 | 1 | |
| | Yes | 19 | 13 | 5.6(3.0-10.3) | 3.1(1.5-6.7) | **0.003** |
| Respiratory rate (breaths per minute) | 12-18 | 18 | 4 | 1 | 1 | |
| | >18 | 371 | 59 | 1.1(0.3-2.6) | 1.5(0.4- 5.4) | 0.562 |
| Antiplatelet | No | 162 | 44 | 1 | 1 | |
| | Yes | 227 | 19 | 0.3(0.2-0.6) | 0.5(0.3-0.9) | **0.014** |

Center, Ethiopia [34], and Felege Hiwot Hospital, Ethiopia [13]. It could be associated with aging, which makes blood walls less elastic and more vulnerable to injury. Fatty deposits may accumulate in the arteries as a result, increasing the risk of obstructions and stroke death [35]. Furthermore, other comorbid diseases that are linked to death are more common in older people [34].

Hypertension was positively associated with mortality in this study. Patients with hypertension were more likely to die from a stroke than those without the condition. According to research done in Sierra Leone [30], in Debre Markos, Ethiopia [19], at Jimma University Medical, Ethiopia [18], and at Ayder Comprehensive Specialized Hospital, Ethiopia [27]. Because hypertension raises the danger of blood clots that might block the passage of blood to the brain, it has a high stroke fatality rate. Additionally, it can harm and weaken blood vessel walls, increasing their vulnerability to rupture [36]. Uncontrolled hypertension might cause damage to other vital organs such as the heart and kidneys, which can further increase the risk of stroke mortality over time [37].

In stroke patients, aspirational pneumonia dramatically raised the risk of death. These results are consistent with research done in Sierra Leone [30], in Lusaka, Zambia [38], in Tanzania [39], at Tibebe Ghion and Felege Hiwot hospitals in Amhara, Ethiopia [40], and at Jimma University Medical Center, Ethiopia [41]. Aspirational pneumonia can lead to several respiratory problems, including acute respiratory distress syndrome and respiratory failure, which can worsen the course of treatment for stroke patients and raise their chance of dying [42].

Lower GCS was significantly associated with mortality among stroke patients. The Glasgow Coma Scale for moderate impairment [9–12] (AHR = 2.2) and severe impairment [3–8] (AHR = 2.4) were found to be statistically significant predictors of in-hospital mortality in a study carried out at Debre Markos Comprehensive Specialized Hospital in Ethiopia [28]. This finding is also in line with the finding of a study conducted in Lusaka, Zambia [38], at Tibebe Ghion and Felege Hiwot hospitals in Amhara, Ethiopia [40], Jimma University Medical Center, Ethiopia [41], and at Saint Paul's Hospital Millennium Medical College, Ethiopia [32]. A lower GCS indicates a more severe level of neurological impairment, which is often associated with more extensive brain damage that is directly interrelated with the severity of stroke and risk of mortality [43].

Patients with heart failure were more likely to die from a stroke than those without the condition. This finding is consistent with the findings of a study conducted in Tanzania [39] and at Tingandogo University Hospital in Burkina Faso [44]. This may be due to the elevated risk of mortality from cardiovascular disease, which in turn increases the risk of mortality from cerebrovascular disease. Due to the reduced cardiac output, reduced blood flow to the brain, and additional side effects, such as pulmonary edema, heart failure, it carries a high risk of death [33].

Stroke patients who experienced complications had a greater chance of dying than those who did not. According to research conducted at the Felege Hiwot Referral Hospital, Tibebe Gion Comprehensive Specialized Hospital, and University of Gondar Teaching Hospital, the absence of problems at admission was found to be a factor affecting the 28-day death rate [31]. Complications are indicative of a greater level of physiological disruption and impaired organ function, which might result in adverse consequences [40].

The incidence of mortality due to stroke was greater among individuals with hospital-acquired infections in comparison to those without such infections. This observation corresponds with the findings of a multicenter prospective cohort study involving Lebanese stroke patients, which indicated that the occurrence of infectious complications served as a predictor of mortality at 1 month (HR = 4.2, p = 0.013) and overall mortality (HR = 3.0, p = 0.007, respectively) [45]. Stroke has the potential to impair the patient's immune system, rendering them more susceptible to infections [46]. Hospital-acquired infections may contribute to the onset of sepsis, a critical and potentially fatal condition [47].

Furthermore, the risk of mortality was 50.0% lower among stroke patients who utilized anti-platelet medications in comparison to those who did not utilize anti-platelet medications. Antiplatelet medications such as aspirin or clopidogrel hinder the development of blood clots [48]. Antiplatelet therapy may decrease the extent of brain damage caused by ischemia and simultaneously diminish the probability of early recurrent ischemic stroke, thereby lowering the likelihood of early death and enhancing treatment outcomes [49]. Antiplatelet medications have been demonstrated to possess cardioprotective properties, diminishing the risk of heart failure and other cardiovascular issues that can lead to mortality in stroke patients [50].

Prehospital care-seeking delays are common among patients with stroke, especially in low-income countries. In this study, the average onset-to-door time is 36 hours. Patients who live farther from hospitals, have a lower level of education, diabetes, hyperlipidemia, or a history of stroke are more likely to experience delays [51]. According to a study conducted at Yekatit-12 Hospital Medical College, Ethiopia, age, place of residence, health insurance, and stroke onset time significantly influence the timeliness of seeking medical care [52].

## Strengths and limitations of the study

This study has clarified the current situation of unfavorable treatment outcomes and predictors of mortality among patients with stroke at public hospitals in the Harari region of East Ethiopia. Conducted over a 60-month follow-up period, it effectively highlights the long-term impacts of stroke treatment. The research was carried out at two sites with an adequate sample size, enabling a reflection of the regional burden of stroke and supporting the potential for generalizations. However, since the study is retrospective, which had problems with being unfinished/incomplete, even losing patient medical information, and some important factors that might have a significant association with stroke mortality (road conditions, distances between medical centers, public awareness, occupation, substance use, physical exercise, and educational

status) could not be found on the medical cards and were not assessed. This could underestimate the findings and reduce the statistical power of the study. The study only included public hospitals, and 22.6% of cases are unclassified, which could introduce a bias. Furthermore, the study may have overestimated the rate of mortality due to stroke by assuming that stroke was the exclusive cause of all cases of mortality.

## Conclusion

The mortality rate of stroke in this study was comparable to that of other studies in Ethiopia. Having hypertension, heart failure, a lower GCS, having complications, aspiration pneumonia, and hospital-acquired infection increased the hazards of mortality among stroke patients, whereas taking antiplatelet drugs reduced the hazards of mortality among stroke patients.

Based on the study's findings, we recommend that health professionals prioritize the care of stroke patients with hypertension, heart failure, lower Glasgow Coma Scale scores, and complications such as aspiration pneumonia and hospital-acquired infections. Administrators at HFCSH and JGH should ensure adherence to secondary prevention strategies, including the use of antiplatelet and anticoagulant medications, as well as treatments for managing risk factors like hypertension. Additionally, researchers are encouraged to conduct further prospective follow-up studies to evaluate the incidence and potential predictors of mortality among stroke patients.

## Supporting information

**S1 Data. This is the data set of a study of mortality rate and predictors among stroke patients.**
(XLSX)

## Acknowledgments

We acknowledged Haramaya University College of Health and Medical Sciences Institutional Health Research Ethical Review Committee for giving ethical clearance. We would like to thank Hiwot Fana Comprehensive Specialized University Hospital and Jugal General Hospital administrative bodies and the card room workers for their cooperation and permission to conduct the study. We would also like to thank the data collectors and supervisors for their commitment during data collection.

## Author contributions

**Conceptualization:** Alemayehu Tesfaye, Lemma Demissie Regassa, Assefa Tola.

**Data curation:** Alemayehu Tesfaye, Lemma Demissie Regassa, Nano Belema Areda, Assefa Tola.

**Formal analysis:** Alemayehu Tesfaye, Lemma Demissie Regassa, Birhanu Shegene, Assefa Tola.

**Funding acquisition:** Alemayehu Tesfaye.

**Investigation:** Alemayehu Tesfaye.

**Methodology:** Alemayehu Tesfaye, Lemma Demissie Regassa, Birhanu Shegene, Nano Belema Areda, Assefa Tola.

**Project administration:** Alemayehu Tesfaye.

**Resources:** Alemayehu Tesfaye.

**Software:** Alemayehu Tesfaye.

**Supervision:** Alemayehu Tesfaye, Lemma Demissie Regassa, Birhanu Shegene, Assefa Tola.

**Validation:** Alemayehu Tesfaye, Lemma Demissie Regassa, Birhanu Shegene, Nano Belema Areda, Assefa Tola.

**Visualization:** Alemayehu Tesfaye, Lemma Demissie Regassa.

**Writing – original draft:** Alemayehu Tesfaye.

**Writing – review & editing:** Alemayehu Tesfaye, Lemma Demissie Regassa, Birhanu Shegene, Nano Belema Areda, Assefa Tola.

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
