## [Decision Letter · Decision Letter 0]

8 Apr 2025

PGPH-D-25-00374

Mortality rate and predictors for stroke patients in the public hospital of Harari region, Eastern Ethiopia

Dear Dr. Tesfaye,

Thank you for submitting your manuscript to PLOS Global Public Health. After careful consideration, we feel that it has merit but does not fully meet PLOS Global Public Health’s publication criteria as it currently stands. Therefore, we invite you to submit a revised version of the manuscript that addresses the points raised during the review process.

Please note that we have only been able to secure a single reviewer to assess your manuscript. We are issuing a decision on your manuscript at this point to prevent further delays in the evaluation of your manuscript. Please be aware that the editor who handles your revised manuscript might find it necessary to invite additional reviewers to assess this work once the revised manuscript is submitted. However, we will aim to proceed on the basis of this single review if possible. 

We look forward to receiving your revised manuscript.

Kind regards,

Joanna Tindall, PhD

Staff Editor

Journal Requirements:

1. "We noticed you have some minor occurrence of overlapping text with the following previous publication(s), which needs to be addressed: https://bmccardiovascdisord.biomedcentral.com/articles/10.1186/s12872-024-04106-4 In your revision ensure you cite all your sources (including your own works), and quote or rephrase any duplicated text outside the methods section. Further consideration is dependent on these concerns being addressed." 2. We ask that a manuscript source file is provided at Revision. Please upload your manuscript file as a .doc, .docx, .rtf or .tex. 3. In the online submission form, you indicated that Data upon which the result is based will be available from the corresponding author upon request.  All PLOS journals now require all data underlying the findings described in their manuscript to be freely available to other researchers, either 1. In a public repository, 2. Within the manuscript itself, or 3. Uploaded as supplementary information. This policy applies to all data except where public deposition would breach compliance with the protocol approved by your research ethics board. If your data cannot be made publicly available for ethical or legal reasons (e.g., public availability would compromise patient privacy), please explain your reasons by return email and your exemption request will be escalated to the editor for approval. Your exemption request will be handled independently and will not hold up the peer review process, but will need to be resolved should your manuscript be accepted for publication. One of the Editorial team will then be in touch if there are any issues.

Additional Editor Comments (if provided):

Reviewers' comments:

Reviewer's Responses to Questions

**Comments to the Author**

1. Does this manuscript meet PLOS Global Public Health’s publication criteria? Is the manuscript technically sound, and do the data support the conclusions? The manuscript must describe methodologically and ethically rigorous research with conclusions that are appropriately drawn based on the data presented.

Reviewer #1: Partly

2. Has the statistical analysis been performed appropriately and rigorously?

Reviewer #1: Yes

3. Have the authors made all data underlying the findings in their manuscript fully available (please refer to the Data Availability Statement at the start of the manuscript PDF file)?

Reviewer #1: Yes

4. Is the manuscript presented in an intelligible fashion and written in standard English?

Reviewer #1: Yes

5. Review Comments to the Author

Reviewer #1: In their work, the authors present the epidemiology of stroke, treatment outcomes, and predictors of stroke-related mortality in Ethiopia.

The results of such a study may be useful in assessing the epidemiology of cerebrovascular diseases and in developing healthcare strategies in these regions.

However, the paper contains several shortcomings that should be addressed before publication in a journal.

Comments:

It is difficult to agree with the statement that "Although infections have received relatively little attention in stroke research" because the impact of infections on stroke risk and progression has been studied for many years. One of the first researchers in this area was Pierre Marie in 1885, followed by Zygmunt Freud, who—before becoming a distinguished psychiatrist and psychoanalyst—practiced as a neurologist in Vienna. In Freud’s research, infections preceded the onset of stroke in up to one-third of cases. Later, particularly in the 1990s, further studies were conducted on infections such as Helicobacter pylori and Chlamydia pneumoniae. See, for example, Mendall et al. (1965), Whinup et al. (1996), and Markus & Mendall (1998). More recently, additional studies have been published, such as in Cerebrovasc Dis Extra (2025 Mar 11:1-19) and Emsley HC, Hopkins SJ. "Acute Ischaemic Stroke and Infection: Recent and Emerging Concepts" (Lancet Neurol. 2008;7(4):341-53). Another relevant study is Elkind MS et al. "Hospitalization for infection and risk of acute ischemic stroke: the Cardiovascular Health Study" (Stroke. 2011;42(7):1851-6), which identifies infections as a risk factor for stroke.

In these studies, stroke was most commonly preceded by respiratory infections, which generally aligns with the authors' findings, although this knowledge is secondary to earlier research.

Next Comments:

• The observation period was quite long, spanning from July 1, 2019, to June 30, 2023. Unfortunately, the paper lacks clear information on the total number of patients admitted to all analyzed hospitals during this period. The authors only provide data on a pre-selected group of patients, which can introduce methodological bias, especially given the presence of numerous confounding factors that hinder an accurate assessment of the situation. However, based on the presented data, the authors had probably (? See above mentioned lack of clear information) access to well over 4,000 patients, which is a substantial number allowing for robust statistical conclusions.

• Maybe, it might have been better to conduct a longitudinal study with consecutive cases over a shorter period, including all patients? The authors should more thoroughly explain why they chose their particular study methodology.

• Moreover, the time from symptom onset to hospital admission was notably long—36 hours. The authors should justify this delay, which is likely due to factors such as poor road conditions, long distances between medical centers, and low public awareness. However, it would be advisable for the authors to explicitly discuss these issues. On the other hand, such a long delay precludes effective acute-phase treatment.

• Additionally, the authors report that over 35% of patients had hemorrhagic stroke, which is a very high proportion although is known that studies have shown that individuals of African descent have a 2-3 times higher risk of experiencing a hemorrhagic stroke compared to the White population, it is also requires explanation, for readers not very familiar with stroke epidemiology

6. PLOS authors have the option to publish the peer review history of their article (what does this mean?). If published, this will include your full peer review and any attached files.

**Do you want your identity to be public for this peer review?** For information about this choice, including consent withdrawal, please see our Privacy Policy.

Reviewer #1: **Yes: **Radosław Kaźmierski, MD, PhD

---

## [Decision Letter · Decision Letter 1]

29 May 2025

PGPH-D-25-00374R1

Mortality rate and predictors for stroke patients in the public hospital of Harari region, Eastern Ethiopia

Dear Dr. Tesfaye,

Thank you for submitting your manuscript to PLOS Global Public Health. After careful consideration, we feel that it has merit but does not fully meet PLOS Global Public Health’s publication criteria as it currently stands. Therefore, we invite you to submit a revised version of the manuscript that addresses the points raised during the review process.

We look forward to receiving your revised manuscript.

Kind regards,

Elliot Koranteng Tannor, MBChB, FWACP, MPhil(Neph), Cert Neph(SA), MBA

Academic Editor

Journal Requirements:

Additional Editor Comments (if provided):

Reviewers' comments:

Reviewer's Responses to Questions

**Comments to the Author**

1. If the authors have adequately addressed your comments raised in a previous round of review and you feel that this manuscript is now acceptable for publication, you may indicate that here to bypass the “Comments to the Author” section, enter your conflict of interest statement in the “Confidential to Editor” section, and submit your "Accept" recommendation.

Reviewer #1: All comments have been addressed

Reviewer #2: (No Response)

2. Does this manuscript meet PLOS Global Public Health’s publication criteria? Is the manuscript technically sound, and do the data support the conclusions? The manuscript must describe methodologically and ethically rigorous research with conclusions that are appropriately drawn based on the data presented.

Reviewer #1: Yes

Reviewer #2: Partly

3. Has the statistical analysis been performed appropriately and rigorously?

Reviewer #1: Yes

Reviewer #2: Yes

4. Have the authors made all data underlying the findings in their manuscript fully available (please refer to the Data Availability Statement at the start of the manuscript PDF file)?

Reviewer #1: Yes

Reviewer #2: Yes

5. Is the manuscript presented in an intelligible fashion and written in standard English?

Reviewer #1: Yes

Reviewer #2: No

6. Review Comments to the Author

Reviewer #1: no further comments

Reviewer #2: Title: The title of the manuscript needs to be modified.

• The title of the manuscript, ‘Mortality rate and predictors for stroke patients in the public hospital of Harari region, Eastern Ethiopia’, gives the impression that the authors are speaking of a single public hospital, which is the only public hospital in Harari. The authors should therefore consider modifying the title to ‘Mortality rate and predictors among stroke patients in public hospitals in the Harari region, Eastern Ethiopia’ or ‘Mortality rate and its predictors among stroke patients in public hospitals in the Harari region, Eastern Ethiopia’

Abstract

• Background: the background of the abstract neither reads well nor seems to convey the contextual significance that the authors appear to want to communicate. The authors need to rewrite to improve the clarity of thought and expression.

• Results: The authors need to clarify the criteria used to determine improvement in patients (e.g., functional outcome scale, NIHSS, or another metric). This should be included in the methods section of the main text.

• Kindly replace 'left against medical advice' with the more standard term 'discharged against medical advice.' Ensure consistency regarding the use of 'mortality rate' instead of 'death rate'.

• Conclusion: the conclusion ‘In-hospital mortality was observed in one of seven stroke patients treated at public hospitals.’ It is out of place and does not align with the results provided in the abstract.

Main text

• Introduction: The introduction should be succinct and straight to the point. The authors should concentrate on the scientific premise, evidence gap, healthcare environment, and regional burden that support the study. The definition of stroke, for instance, is unnecessary. The same applies to the paragraph on management options for stroke in a broader context rather than what pertains in their region. The first sentence of paragraph 3 of the introduction ‘The fact that 94% of ischemic stroke patients survived and 6% passed away is remarkable. Hemorrhagic stroke patients, on the other hand, had a survival rate of 88%, while 12% died (6).’ is out of place and should be deleted. I recommend that the authors make the introduction more concise.

Methods: The methods section requires a bit more detail.

• Populations and Eligibility: The exclusion criteria are very scanty. For example, were patients with incomplete medical records excluded? If so, this should be included as an exclusion criterion. How did you deal with missing data? Were the retrospective records retrieved from electronic health records, or paper charts, or were there existing stroke registries?

• What proportion of patients what neuroimaging confirmed strokes? The statement ‘whereas stroke patients whose primary diagnosis was not a stroke (e.g., transient ischemic attacks without stroke confirmation)’ is poorly structured and seems to suggest all patients had neuroimaging. Currently the diagnosis of transient ischemic attacks is tissue based and requires the use of neuroimaging to rule out strokes.

• The authors also state that ‘transfer-in patients within the follow-up period were excluded from the study’. Considering the hospitals being studied are tertiary facilities, I expect a significant proportion of the patients to be referred from lower-level facilities. Can the authors explain why they were excluded?

• Sampling methods. The authors need to clarify how the sampling frame was constructed and whether any pre-screening was done. How was the list of eligible patients compiled? Were exclusion criteria applied before this apparent random selection of participants?

• Also, it would be helpful to include a flow diagram indicating the total number of records screened. The proportion excluded and reasons for exclusion (the number excluded for incomplete records, transfer-in/ referred from other facilities, etc. and the final number of patients included.

• Regarding the sample size calculation for the second objective, specify assumed proportions or effect size. At least one key factor used.

• Variables: how was raised ICP? Was it via invasive ICP monitoring? For stroke patients, it is surprising to see heart failure contributing more than other comorbidities like diabetes. This is appropriately highlighted in the discussion section. It would be helpful to indicate how these heart failures were diagnosed in the methods, to clarify the possibility of overestimation. Did the patients have echocardiograms done? Also, why is dyslipidemia not included in the comorbidities?

• In retrospective chart reviews, it is impractical to say that "Authors had no access to information that could identify individual participants during or after data collection." This statement should be removed or revised.

Results:

• The authors state that there was a total of 477 samples which ambiguous. Reword to reflect the fact that a total of 477 records were screened. Also, screen failures should not be included in the total sample size. If patients with incomplete records were excluded, report this both in the methods and in the flow of the results.

• Further classify hemorrhagic stroke into subtypes (e.g., intracerebral hemorrhage, subarachnoid hemorrhage). Provide a more detailed breakdown of stroke-related complications.

• Do the authors have information regarding the length of stay of patients. Can this be incuded?

• There is no data included on the interventions received by patients such as thrombolysis, thrombectomy, EVD, aneurysmal clipping/ coiling etc. how did the treatments received by patients affect mortality?

• Kindly provide the appropriate labels for the Kaplan-Meier curve to enhance interpretability. The y-axis should be Labelled, and the x-axis should indicate follow-up time or time to event, not “analysis time.” Also, it would be helpful to stratify the curves by important factors such as aspiration pneumonia, etc.

Limitations:

• The study only included public hospitals, which could introduce significant bias. This should be included in the limitation.

• 22.6% of cases are unclassified. This is a significant limitation.

Response to previous reviewer comments

The authors have failed to adequately address some of the concerns raised by the previous authors. The authors are urged to see the review process as an endeavor to improve the manuscript's quality. Most of the recommendations from the previous reviewers should be incorporated to strengthen the paper.

1. The revision of the sentence ‘Although infections have received relatively little attention in stroke research’ to ‘Infection is associated with a risk of stroke’ remains out of context. A more accurate phrasing would be: ‘Stroke is associated with an increased risk of post-stroke infections, which can influence outcomes.

2. As was rightly pointed out by the previous reviewers, the average onset-to-door time of 36 hours is clinically significant and warrants discussion. . Although not a primary objective, this is a key finding and should be acknowledged and discussed using insights from relevant regional studies. The delay may reflect systemic issues such as poor road infrastructure, long distances to care, or low public awareness.

7. PLOS authors have the option to publish the peer review history of their article (what does this mean?). If published, this will include your full peer review and any attached files.

**Do you want your identity to be public for this peer review?** For information about this choice, including consent withdrawal, please see our Privacy Policy.

Reviewer #1: **Yes: **prof. Radosław Kaźmierski, MD, PhD

Reviewer #2: No

---

## [Editor Report · Decision Letter 2]

9 Jun 2025

PGPH-D-25-00374R2

Mortality rate and its predictors among stroke patients in the public hospitals in Harari region, Eastern Ethiopia

Dear Dr. Tesfaye,

Thank you for submitting your manuscript to PLOS Global Public Health. After careful consideration, we feel that it has merit but does not fully meet PLOS Global Public Health’s publication criteria as it currently stands. Therefore, we invite you to submit a revised version of the manuscript that addresses the points raised during the review process.

We look forward to receiving your revised manuscript.

Kind regards,

Elliot Koranteng Tannor, MBChB, FWACP, MPhil(Neph), Cert Neph(SA), MBA

Academic Editor

Journal Requirements:

Additional Editor Comments .

Find attached the reviewers comments for your perusal.
---

## [Editor Report · Decision Letter 3]

18 Jun 2025

PGPH-D-25-00374R3

Mortality rate and its predictors among stroke patients in the public hospitals in Harari region, Eastern Ethiopia

Dear Dr. Tesfaye,

Thank you for submitting your manuscript to PLOS Global Public Health. After careful consideration, we feel that it has merit but does not fully meet PLOS Global Public Health’s publication criteria as it currently stands. Therefore, we invite you to submit a revised version of the manuscript that addresses the points raised during the review process.

Please ensure that your decision is justified on PLOS Global Public Health’s publication criteria and not, for example, on novelty or perceived impact.

We look forward to receiving your revised manuscript.

Kind regards,

Elliot Koranteng Tannor, MBChB, FWACP, MPhil(Neph), Cert Neph(SA), MBA

Academic Editor

Journal Requirements:

1. We noticed you have some minor occurrence of overlapping text with the following previous publication(s), which needs to be addressed:

https://bmccardiovascdisord.biomedcentral.com/articles/10.1186/s12872-024-04106-4

In your revision ensure you cite all your sources (including your own works), and quote or rephrase any duplicated text outside the methods section. Further consideration is dependent on these concerns being addressed.

Additional Editor Comments (if provided):

Reviewers' comments:

Reviewer 2 Comments

Title: The title of the manuscript needs to be modified.

• The title of the manuscript, ‘Mortality rate and predictors for stroke patients in the public hospital of Harari region, Eastern Ethiopia’, gives the impression that the authors are speaking of a single public hospital, which is the only public hospital in Harari. The authors should therefore consider modifying the title to ‘Mortality rate and predictors among stroke patients in public hospitals in the Harari region, Eastern Ethiopia’ or ‘Mortality rate and its predictors among stroke patients in public hospitals in the Harari region, Eastern Ethiopia’

Abstract

• Background: the background of the abstract neither reads well nor seems to convey the contextual significance that the authors appear to want to communicate. The authors need to rewrite to improve the clarity of thought and expression.

• Results: The authors need to clarify the criteria used to determine improvement in patients (e.g., functional outcome scale, NIHSS, or another metric). This should be included in the methods section of the main text.

• Kindly replace 'left against medical advice' with the more standard term 'discharged against medical advice.' Ensure consistency regarding the use of 'mortality rate' instead of 'death rate'.

• Conclusion: the conclusion ‘In-hospital mortality was observed in one of seven stroke patients treated at public hospitals.’ It is out of place and does not align with the results provided in the abstract.

Main text

• Introduction: The introduction should be succinct and straight to the point. The authors should concentrate on the scientific premise, evidence gap, healthcare environment, and regional burden that support the study. The definition of stroke, for instance, is unnecessary. The same applies to the paragraph on management options for stroke in a broader context rather than what pertains in their region. The first sentence of paragraph 3 of the introduction ‘The fact that 94% of ischemic stroke patients survived and 6% passed away is remarkable. Hemorrhagic stroke patients, on the other hand, had a survival rate of 88%, while 12% died (6).’ is out of place and should be deleted. I recommend that the authors make the introduction more concise.

Methods: The methods section requires a bit more detail.

• Populations and Eligibility: The exclusion criteria are very scanty. For example, were patients with incomplete medical records excluded? If so, this should be included as an exclusion criterion. How did you deal with missing data? Were the retrospective records retrieved from electronic health records, or paper charts, or were there existing stroke registries?

• What proportion of patients what neuroimaging confirmed strokes? The statement ‘whereas stroke patients whose primary diagnosis was not a stroke (e.g., transient ischemic attacks without stroke confirmation)’ is poorly structured and seems to suggest all patients had neuroimaging. Currently the diagnosis of transient ischemic attacks is tissue based and requires the use of neuroimaging to rule out strokes.

• The authors also state that ‘transfer-in patients within the follow-up period were excluded from the study’. Considering the hospitals being studied are tertiary facilities, I expect a significant proportion of the patients to be referred from lower-level facilities. Can the authors explain why they were excluded?

• Sampling methods. The authors need to clarify how the sampling frame was constructed and whether any pre-screening was done. How was the list of eligible patients compiled? Were exclusion criteria applied before this apparent random selection of participants?

• Also, it would be helpful to include a flow diagram indicating the total number of records screened. The proportion excluded and reasons for exclusion (the number excluded for incomplete records, transfer-in/ referred from other facilities, etc. and the final number of patients included.

• Regarding the sample size calculation for the second objective, specify assumed proportions or effect size. At least one key factor used.

• Variables: how was raised ICP? Was it via invasive ICP monitoring? For stroke patients, it is surprising to see heart failure contributing more than other comorbidities like diabetes. This is appropriately highlighted in the discussion section. It would be helpful to indicate how these heart failures were diagnosed in the methods, to clarify the possibility of overestimation. Did the patients have echocardiograms done? Also, why is dyslipidemia not included in the comorbidities?

• In retrospective chart reviews, it is impractical to say that "Authors had no access to information that could identify individual participants during or after data collection." This statement should be removed or revised.

Results:

• The authors state that there was a total of 477 samples which ambiguous. Reword to reflect the fact that a total of 477 records were screened. Also, screen failures should not be included in the total sample size. If patients with incomplete records were excluded, report this both in the methods and in the flow of the results.

• Further classify hemorrhagic stroke into subtypes (e.g., intracerebral hemorrhage, subarachnoid hemorrhage). Provide a more detailed breakdown of stroke-related complications.

• Do the authors have information regarding the length of stay of patients. Can this be incuded?

• There is no data included on the interventions received by patients such as thrombolysis, thrombectomy, EVD, aneurysmal clipping/ coiling etc. how did the treatments received by patients affect mortality?

• Kindly provide the appropriate labels for the Kaplan-Meier curve to enhance interpretability. The y-axis should be Labelled, and the x-axis should indicate follow-up time or time to event, not “analysis time.” Also, it would be helpful to stratify the curves by important factors such as aspiration pneumonia, etc.

Limitations:

• The study only included public hospitals, which could introduce significant bias. This should be included in the limitation.

• 22.6% of cases are unclassified. This is a significant limitation.

Response to previous reviewer comments

The authors have failed to adequately address some of the concerns raised by the previous authors. The authors are urged to see the review process as an endeavor to improve the manuscript's quality. Most of the recommendations from the previous reviewers should be incorporated to strengthen the paper.

1. The revision of the sentence ‘Although infections have received relatively little attention in stroke research’ to ‘Infection is associated with a risk of stroke’ remains out of context. A more accurate phrasing would be: ‘Stroke is associated with an increased risk of post-stroke infections, which can influence outcomes.

2. As was rightly pointed out by the previous reviewers, the average onset-to-door time of 36 hours is clinically significant and warrants discussion. . Although not a primary objective, this is a key finding and should be acknowledged and discussed using insights from relevant regional studies. The delay may reflect systemic issues such as poor road infrastructure, long distances to care, or low public awareness

---

## [Decision Letter · Decision Letter 4]

20 Aug 2025

Mortality rate and predictors among stroke patients in the public hospitals in Harari region, Eastern Ethiopia

PGPH-D-25-00374R4

Dear Mr. Tesfaye,

We are pleased to inform you that your manuscript 'Mortality rate and predictors among stroke patients in the public hospitals in Harari region, Eastern Ethiopia' has been provisionally accepted for publication in PLOS Global Public Health.

Best regards,

Elliot Koranteng Tannor, MBChB, FWACP, MPhil(Neph), Cert Neph(SA), MBA

Academic Editor

Reviewer Comments (if any, and for reference):

Reviewer's Responses to Questions

**Comments to the Author**

1. If the authors have adequately addressed your comments raised in a previous round of review and you feel that this manuscript is now acceptable for publication, you may indicate that here to bypass the “Comments to the Author” section, enter your conflict of interest statement in the “Confidential to Editor” section, and submit your "Accept" recommendation.

Reviewer #2: All comments have been addressed

2. Does this manuscript meet PLOS Global Public Health’s publication criteria? Is the manuscript technically sound, and do the data support the conclusions? The manuscript must describe methodologically and ethically rigorous research with conclusions that are appropriately drawn based on the data presented.

Reviewer #2: Yes

3. Has the statistical analysis been performed appropriately and rigorously?

Reviewer #2: Yes

4. Have the authors made all data underlying the findings in their manuscript fully available (please refer to the Data Availability Statement at the start of the manuscript PDF file)?

Reviewer #2: Yes

5. Is the manuscript presented in an intelligible fashion and written in standard English?

Reviewer #2: Yes

6. Review Comments to the Author

Reviewer #2: N/A

7. PLOS authors have the option to publish the peer review history of their article (what does this mean?). If published, this will include your full peer review and any attached files.

**Do you want your identity to be public for this peer review?** For information about this choice, including consent withdrawal, please see our Privacy Policy.

Reviewer #2: No
